# Improving the Generalization of Visual Navigation Policies using Invariance Regularization

**Michel Aractingi** [1]   **Christopher Dance** [1]   **Julien Perez** [1]   **Tomi Silander** [1]

## Abstract

Training agents to operate in one environment often yields overfitted models that are unable to *generalize* to the changes in that environment. However, due to the numerous variations that can occur in the real-world, the agent is often required to be robust in order to be useful. This has not been the case for agents trained with *reinforcement learning (RL)* algorithms. In this paper, we investigate the overfitting of RL agents to the training environments in visual navigation tasks. Our experiments show that deep RL agents can overfit even when trained on multiple environments simultaneously. We propose a regularization method which combines RL with supervised learning methods by adding a term to the RL objective that would encourage the *invariance* of a policy to variations in the observations that ought not to affect the action taken. The results of this method, called *Invariance Regularization*, show an improvement in the generalization of policies to environments not seen during training. The experimentation is done on the VizDoom environment which contains hundreds of textures, so allowing us to investigate generalization to changes in the visual observation.

## 1. Introduction

Learning control policies from high-dimensional sensory input has been gaining more traction lately due to the popularity of deep reinforcement learning (DRL) (Mnih et al., 2015; Levine et al., 2015), which enables learning the perception and control modules simultaneously. However, most of the work done in RL chooses to perform the evaluation of the learned policies in the same environment in which training occurred (Cobbe et al., 2018).

[1]NAVER LABS Europe, Grenoble, France. Correspondence to: *firstname lastname* <firstname.lastname@naverlabs.com>.

*Reinforcement Learning for Real Life (RL4RealLife) Workshop in the $36^{th}$ International Conference on Machine Learning*, Long Beach, California, USA, 2019. Copyright 2019 by the author(s).

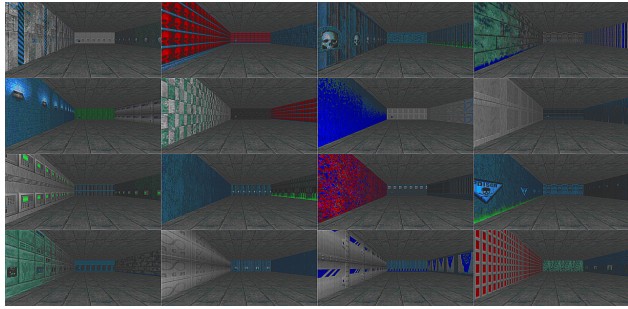

*Figure 1.* The figure shows how environments may differ in their visual aspects, like textures of the surfaces. The textures provide a differentiator for each environment, where without them the environments would have shared the same state space.

Using the same environments to train and test agents does not give any insight about the generalization abilities of the learned policy. There could be a number of changes in the environment at test time that would degrade the agent's performance. Variations could appear in the visual aspects that determine the agent's observation, the physical structure that determines the agent's state and even some aspects that are related to the agent's goal (Figure 1). For example, different observations of the same room are encountered at different times of the day (different lighting conditions). New obstacles could be present. Levels of a game could be different, yet playing a few levels should often be enough to figure out how to play the rest. Such variations might result in a *new* environment where the control model that defined the training environment has changed. A robust policy should generalize from its experience and perform the same skills in the presence of these variations.

DRL agents have been notorious for overfitting to their training environments (Cobbe et al., 2018). An agent could have drastically different performance on testing environments even if it manages to maximize the reward during training (Zhang et al., 2018). Supervised learning algorithms have been shown to have some generalization guarantees when adding proper regularization to the

training set. However, these guarantees are weakened in reinforcement learning algorithms where the source of the data is not *i.i.d.* In order to make use of the progress of DRL algorithms in practice we need policies that are robust to possible changes in the sensory inputs, surrounding structure and even some aspects of the task.

In this paper we study the notion of generalization that is appropriate for visual navigation control policies that are learned with DRL. We present: (1) a study of the generalization of visual control policies to certain changes in the underlying dynamical system; (2) an alternative training method that combines DRL with supervised learning, thus using DRL to learn a controller while leveraging the generalization properties of supervised learning. In our experiments we use the VizDoom platform (Kempka et al., 2016) which is easily customizable and enables the generation of numerous variants of a given environment.

## 2. Preliminaries

Visual navigation for mobile robots combines the domains of vision and control. Navigation can be described as finding a suitable and safe path between a starting state and a goal state (Bonin-Font et al., 2008). Classical approaches split the problem to a sequence of sub-tasks, such as map construction, localization, planning and path following (Bonin-Font et al., 2008). However, each sub-task requires some hand-engineering that is specific to the environment and task which makes it hard to adapt it to different scenarios without performing some tuning. Deep learning approaches enable the use of highly non-linear classifiers that can adapt their inner representations to learn to robustly solve complicated tasks (Goodfellow et al., 2016).

In this work, we use reinforcement learning algorithms coupled with deep learning approaches to solve the task of navigating an agent towards a goal object using only its visual observations as input. The field of view of the agent is limited, *i.e.*, it does not observe the full environment, and we do not provide an explicit map of the environment to that agent.

### 2.1. Problem Statement

We model the problem of visual navigation as a partially observed Markov decision process (POMDP) (Spaan, 2012). A POMDP is given by a tuple

$$\mathcal{P} := \langle \mathcal{S}, \mathcal{A}, \Omega, R, T, O, P_0 \rangle,$$

where $\mathcal{S}$ is the set of states, $\mathcal{A}$ is the set of actions and $\Omega$ is the set of observations, all which are assumed to be finite sets. The reward function is $R : \mathcal{S} \times \mathcal{A} \to \mathbb{R}$.

The conditional transition probability mass function is $T : \mathcal{S} \times \mathcal{A} \times \mathcal{S} \to [0, 1]$, with the interpretation that $T(s, a, s') = p(s_{t+1} = s' | s_t = s, a_t = a)$ is the probability that the next state is $s'$ given that the current state is $s$ and that action $a$ is taken. The conditional observation probability mass function is $O : \mathcal{S} \times \mathcal{A} \times \Omega \to [0, 1]$, with the interpretation that $O(s, a, o) = p(o_t = o | s_t = s, a_{t-1} = a)$ is the probability of observing $o$ in state $s$ when the last action taken was $a$, and we allow for a special observation probability $O(s, o) = p(o_0 = o | s_0 = s)$ when in the initial state $s$ and no action has yet been taken. Finally, $P_0$ is the initial state probability mass function, so that $P_0(s) = p(s_0 = s)$ is the probability that the initial state is $s$.

In DRL, we work with a parameterized policy $\pi_\theta(h, a) = p_\theta(a_t = a | h_t = h)$ with parameters $\theta \in \Theta$, giving the probability of taking action $a$ given observation-action history $h_t := (o_0, a_0, o_1, a_1, \ldots, a_{t-1}, o_t)$. The objective is to adjust the parameters $\theta$ to attain a high value for the discounted reward

$$J_\mathcal{P}(\theta) := \mathbb{E}_\mathcal{P}^{\pi_\theta} \left[ \sum_{t=0}^\infty \gamma^t R(s_t, a_t) \right]$$

with discount factor $\gamma \in [0, 1)$. The expectation is over state-observation-action sequences $(s_t, o_t, a_t)_{t=0}^\infty$ where the initial state $s_0$ is drawn from $P_0$ and other elements of such a sequence are drawn from $T, O$ and $\pi_\theta$ respectively (Sutton & Barto, 1998).

Many methods for attempting to approximate optimal policies have been proposed. For instance, *policy gradient methods* perform gradient ascent on estimates of the expected discounted reward. In this work we use the *proximal policy optimization (PPO)* algorithm, which arguably shows relatively robust performance on a wide range of different tasks (Schulman et al., 2017).

### 2.2. Formalizing Generalization

As in classification, we wish to learn from a finite training set but still perform well on previously-unseen examples from a test set. To formalize this, we have a distribution $\mathcal{D}$ over POMDPs, representing multiple environments or tasks, and we sample $n^{\text{train}}$ POMDPs from this distribution $\mathcal{P}_1, \mathcal{P}_2, \ldots, \mathcal{P}_{n^{\text{train}}}$. In the context of navigation, these POMDPs might differ in terms of their observation distributions, perhaps representing views of the same environment at different times of day or year, in terms of their transition distributions, perhaps representing maps with different geometries, or in terms of their reward distributions, perhaps corresponding to the specification of different goal states. Given this sample, we then learn a policy $\pi_\theta$ from a finite collection of

state-observation-action sequences from these POMDPs. In order to have a meaningful common policy across these POMDPs, we require that they have common state, action and observation spaces $\mathcal{S}, \mathcal{A}$ and $\Omega$. By analogy with the notion of generalization risk in classification (Mohri et al., 2018), we say that policy $\pi_\theta$ *generalizes well* if it attains a high value for the expectation of the discounted reward over the full distribution of POMDPs, which we call the *discounted generalization reward*, so that $\mathbb{E}_{\mathcal{P} \sim \mathcal{D}} J_{\mathcal{P}}(\theta)$ is high in some sense. We may interpret 'high' relative to the maximum attainable value for this expectation if it is known to exist, or more generally relative to its supremum $\sup_{\theta \in \Theta} \mathbb{E}_{\mathcal{P} \sim \mathcal{D}} J_{\mathcal{P}}(\theta)$. This is our own terminology as we did not find a semantically-equivalent term in the literature.

It is not hard to see that the discounted generalization reward is actually the discounted reward $J_{\mathcal{P}^{\mathcal{D}}}(\theta)$ of a single larger POMDP $\mathcal{P}^{\mathcal{D}}$, whose state space may however no longer be finite. To see this, let us associate a *unique identifier* $i(\mathcal{P})$ with any POMDP sampled from $\mathcal{D}$ and let $\mathcal{I}^{\mathcal{D}}$ be the set of all such unique identifiers. In the large POMDP $\mathcal{P}^{\mathcal{D}}$, the state space is the Cartesian product $\mathcal{S} \times \mathcal{I}^{\mathcal{D}}$ of the original states and these unique identifiers, but the action and observation spaces are just $\mathcal{A}$ and $\Omega$. The initial state distribution is obtained by first sampling a POMDP $\mathcal{P} \sim \mathcal{D}$ and then sampling $s_{\text{env}} \sim P_0^{\mathcal{P}}$ from that POMDP's initial state distribution. The initial state in the large POMDP is then the concatenation $(s_{\text{env}}, i(\mathcal{P}))$. Transitions, observations and rewards are then made according to the respective distributions and functions of $\mathcal{P}$ but only the first part of the state is involved, the unique identifier $i(\mathcal{P})$ remaining constant. Thus one might succinctly state the problem of generalization in POMDPs as follows: given a distribution $\mathcal{D}$ over POMDPs with common state, action and observation spaces and access to a sample of state-observation-action sequences from a sample of POMDPs drawn from $\mathcal{D}$, choose a policy $\pi_\theta$ that obtains a high value for the discounted reward $J_{\mathcal{P}^{\mathcal{D}}}(\theta)$.

## 3. Related Work

Training in synthetic environments enables the simulation of huge amounts of experience in a span of a few hours. Simulations are convenient to use when training reinforcement learning agents that are often highly sample inefficient (Sutton & Barto, 1998). There is, frequently, a gap between the synthetic world and the real-world, mainly due to the manner in which the simulators depict the real-world dynamics and visual appearances. Often, these simulated worlds capture the richness and noise of the real-world with low-fidelity (Tobin et al., 2017). Many have tried to propose transfer learning techniques to *bridge* the reality gap in order to still make use of fast simulators for training (Taylor & Stone, 2009).

One popular method to bridge the reality gap is by randomizing some aspects of the training environment. This *domain randomization* technique has been shown to be successful for the transfer of grasping policies from simulated training environments to the real-world (Tobin et al., 2017). However, the learned models resulting from that work are not control policies, but perception modules. Previous work has showed some success in transferring the perception module learned in simulation to the real world, but not the controller.

Cobbe et al. (2018) conduct a large scale study on generalization using a new environment, that resembles an arcade game, which they call *CoinRun*. They experiment by training on different background images and different level structures. They test with different regularization strategies and network architectures finding that the RL agent has a surprising tendency to overfit even to large training sets. Zhang et al. (2018) reach a similar conclusion, when learning in grid-world environments, and state that the agents have a tendency to memorize levels of the training set. Unlike Cobbe et al. (2018), however, they argue that the methods that inject stochasticity into the dynamics of the system to prevent memorization, such as sticky actions (Machado et al., 2017) and random initializations (Hausknecht & Stone, 2015), often do not help. In our work we are interested in generalization when navigating under partial observability unlike the fully observable CoinRun or grid-world environments.

Domain adaptation methods have also been used for simulated to real transfer. They allow models trained on a source domain to generalize to a target domain. Bousmalis et al. (2017) train a generative model to adapt the synthetic images of the simulator to appear like the real environment. It was shown to successfully transfer a grasping policy trained in simulation to the real world. However, they do not discuss whether the policy generalizes when variations happen in the target domain.

Another aspect of generalization is the transfer of learned skills to solve different tasks. In other words, generalization to the goal of the trained agent $g$. Achieving different tasks would require the agent to have the ability to maximize different reward functions. Schaul et al. (2015) consider working with value functions that contain the goal $g$ as part of the agent's state. They call them *universal value functions*. The reward will then become a function of a state-action-goal tuple $(s, a, g)$ instead of a classical state-action pair. In the paper, the authors present universal value function approximators (UVFA). A method that attempts to learn a universal value function estimate $V_\theta(s, g)$. They show that UVFA's can generalize for unseen

state-goal pairs in grid-world setup.

Deep reinforcement learning has been used to train control policies. These DRL based methods generally propose to learn motor control commands from raw camera images, thus mapping pixels to commands that control the robot's motors (Levine et al., 2015). DRL algorithms have been used for various navigation tasks such as goal conditioned navigation (Mirowski et al., 2016; Zhu et al., 2016) and *mapless* navigation (Mirowski et al., 2018).

In the next section we will discuss how to explore the effectiveness of domain randomization techniques for the generalization of visual navigation policies.

## 4. Generalization in Visual Control

The motivation behind domain randomization is that it is assumed to be an effective technique to provide a policy that is invariant to the changes that would appear in the observations. We explore the problem of navigating the agent towards a goal object with random noise added to the agent's observations. If the agent is able to perform the task in an environment defined by a POMDP $\mathcal{P}_1$ then it should still be able to perform the task in another POMDP $\mathcal{P}_2$, if certain features $f$ of the environment that are specific to successfully achieving the task exist and are invariant to these variations, i.e., $f(\mathcal{P}_1) = f(\mathcal{P}_2)$. Domain randomization is typically used to train policies that can generalize to variations and noise in the observations. It is done by training on several POMDP's that share the same $\mathcal{S}, \mathcal{A}, \Omega$ spaces, however each POMDP has its own unique identifier which modifies the state, therefore presenting several variations of the observation of the same state.

We present in the experimentation, Section 6.1, a study on domain randomization when added to RL training and the ability of resulting policies to generalize in unseen POMDPs. We want to investigate if the policy does in fact overfit to the training POMDPs and whether we mitigate that overfitting by training the policies on multiple POMDPs. We also discuss the role that invariant channels might play in the success of domain randomization techniques in achieving a policy that is robust to changes in observations.

## 5. Invariance Regularization

In the previous sections, we discussed how overfitting to the training environment can be a big problem in RL. Furthermore, we should be careful not to jump to the conclusion that training on different environments will ensure policies that generalize well to new environments. It is merely an assumption that has been shown to empirically

hold up when used in a supervised learning context. However, we show in this work that this assumption might not hold for reinforcement learning techniques. This is compatible with the findings in Cobbe et al. (2018) and Zhang et al. (2018).

We reason that in order to generalize well, the training objective should include a term that encourages policy generalization. Therefore, putting the weight of the problem of generalizing explicitly in the objective function itself. Formally, a function $h$ of variable $x$ is invariant to a transformation $\phi$ of $x$ if $h(x) = h(\phi(x))$. We can deduce the same definition for the invariance of a policy $\pi$ to changes in the observation given by some transformation $\mathcal{T}$, $\pi(o) = \pi(\mathcal{T}(o))$. We add this regularization penalty term to the RL objective as shown in Equation (1):

$$\max_{\theta} \quad L_{ppo}(O; \pi_{\theta})-$$
$$\frac{\lambda}{NM} \sum_{i=1}^{N} \sum_{j=1}^{M} d(\pi_{\theta}(o_i), \pi_{\theta}(\mathcal{T}_j(o_i))), \quad (1)$$

where $L_{ppo}$ is the PPO objective (Schulman et al., 2017), $\theta$ is the set of parameters that define the policy $\pi_{\theta}$, $d$ is a distance function between the two conditional distributions, and $\lambda$ is a weighting coefficient of the penalty. $O = \{o_1, o_2, ...o_N\}$ is a sequence of $N$ observations.

$\mathcal{T}$ is a transformation of the observations. Given an observation $o$ and a transformation on that observation $\mathcal{T}$ where the transformation still holds the semantic context of the underlying state, but with added visual variations. We can think of the difference between observing a room with observation $o$ and observing the same room with observation $\mathcal{T}(o)$ as the color of the wall for example. Therefore, let us say that we observe $o$ in POMDP $\mathcal{P}$ and observe $\mathcal{T}(o)$ in POMDP $\mathcal{P}^{\mathcal{T}}$ then $f(\mathcal{P}) = f(\mathcal{P}^{\mathcal{T}})$, where $f(\mathcal{P})$ is the set of invariant features of the environment defined by the POMDP $\mathcal{P}$. We further discuss the nature of $\mathcal{T}$ in the experiments section. $M$ is the number of transformations of each observations.

The penalty $d$ in Equation 1 resembles adding a constraint on the PPO objective, where the new objective dictates that the policy should simultaneously obtain a high reward while behaving similarly for the observations $o$ and $\mathcal{T}(o)$. The idea is similar, in spirit, to trust region policy optimization (Schulman et al., 2015) where a penalty term, resembling that which would result from imposing a trust-region constraint, is added to ensure monotonic improvement of the average return with each policy update. We call this method in Equation 1 *invariance regularization* (IR) since the regularization term indicates the invariance of the learned policy to a transformation of given observations.

**Algorithm 1** RL with iterative supervision

Initialize $k_1, k_2, \theta_0, \mathcal{T}_{i=\{1\dots,N\}}, env$
**while** *not converged* **do**
  **for** $i = 1 \dots, k_1$ **do**
    // Train $\pi_\theta$ on $env$ on the RL objective
    $\theta_i \leftarrow \max_\theta L_{ppo}(o^{env}; \pi_{\theta_{i-1}})$
  **end for**
  **for** $j = 1 \dots, k_2$ **do**
    // Train $\pi$ on $env$ and $\mathcal{T}(env)$
    Sample $\{o_t^{env}, \pi_{\theta_{k_1}}(o_t^{env})\}$
    Generate $\{o_t^{\mathcal{T}_i(env)}, \pi_{\theta_{k_1}}(o_t^{\mathcal{T}_i(env)})\}^{i=1\dots N}$
    $\theta_j \leftarrow \min_\theta d(\pi_{\theta_{k_1}}(o_{env}) || \pi_{\theta_{j-1}}(\mathcal{T}_i(o_{env}));$
  **end for**
**end while**
**return** $\pi_\theta$

We propose two ways to solve the RL problem in Equation 1. The first is to directly optimize the full objective by adding the penalty to the original PPO loss. The second method works in an iterative manner, as shown in Algorithm 1, by splitting the training process to two stages of training RL first and then training with supervised learning on the signal from $d(\pi(o), \pi(\mathcal{T}(o)))$ which presents an elegant form that combines reinforcement learning with supervised learning.

In the next section, we will discuss experiments using both methods. Before that, we will describe a study on the effectiveness of domain randomization as a mean to reducing overfitting in DRL agents.

## 6. Experiments

In this section we present the results of two experiments. The first is about training RL with domain randomization. We discuss the ability of the learned policies to generalize to unseen environments when trained on variations of the training environment. The next part presents the results obtained when using the invariance regularization (IR) method, proposed in Section 5, with domain randomization and shows that it improves the success rate considerably.

We performed these experiments because we are interested in the following questions: (1) Does training on environments with random variations (as domain randomization suggests) learn a representation of the invariant $f$ with which the policy can generalize to other environments that share the same invariant features? (2) Can we find a training algorithm that would empirically guarantee finding these invariant features $f$?

### 6.1. Domain Randomization

We leverage the customizability of VizDoom maps (Kempka et al., 2016) with hundreds of unique textures to generate train/test scenarios. The agent is required to reach an object in order to get a reward. We train an actor-critic style agent (Konda & Tsitsiklis, 1999) to solve the task. The network consists of three convolutional layers and 2 fully connected layers, followed by the policy and value function estimator layers. The policy output is a four-dimensional fully-connected layer, where the four dimensions corresponds to four actions; move forward, turn right, turn left and do nothing. The ouput of the policy layer is a log-probability of each action. The value layer is a single unit that predicts the value function. This network architecture was proposed by Mnih et al. (2015). ReLUs are used as the non-linear operations in all layers (Nair & Hinton, 2010). As mentioned, we optimize the PPO objective (Schulman et al., 2017) with a binary reward function (+1 if goal is reached, 0 otherwise) and a discount factor $\gamma = 0.99$.

We generate the variations of the training environment by changing the textures on the surfaces using the numerous textures provided by VizDoom (Kempka et al., 2016). We train agents on a subset of 1, 10, 50, 100 and 500 rooms from the generated environments and test on 50 rooms with textures from a hold-out set which are different from the ones used to generate the training environments. During training we run several agents in parallel to quickly collect observation-action-reward data in multiple environments. Another advantage of this parallelization is the ability to run each agent on a variation of the training environment. Due to hardware limitations, we cannot run one agent for each environment, at least not when we have a large number of training environments, i.e., 100 or 500. Therefore, each agent samples one environment from the training set and runs on it for some $n$ episodes before sampling another one ($n = 25$ episodes). We experiment with different types of visual input; RGB, RGB-D and Grayscale. The number of training iterations is fixed at $5 \times 10^6$ to ensure repeatability of our experiment. The results are therefore potentially pessimistic, and in future work we would like to choose the number of iterations for each network independently so as to maximize generalization performance.

**The role of depth.** Adding a depth channel to the observation plays a significant role in generalization. Depth is invariant to many changes in the visible spectrum of the observations. This might lead the training agent to partly find an invariance in observations in its implicit perception model, which in this case can be as simple as focusing on the depth channel only. Therefore, it was not surprising to see, in Table 1, that the depth agents (RGB-D) generalize

better than the agents without any depth information.

Table 1 shows the success rate of the PPO models with respect to the number of training environments used and the input type (RGB, RGB-D). The results are averaged over 5 seeds, a standard practice in the RL literature today. We notice the superior performance of the agent with depth than the agent without depth. The fact that the RGB agent is not able to generalize well even when exposed to numerous environments tells us that it might not be learning the invariance relating the environments. On the other hand, the RGB-D agents perform well on the testing environments even when the agents are only exposed to 10 random training environments.

Looking at the RGB and RGB-D experiments, the agents trained on 100 and 500 environments generalize worse on average than the ones trained on 10 and 50, which indicates that some agents might be overfitting. This is inspite of the fact that these agents are able to maximize the reward in the training set regardless of the set size. Looking at the *max* statistic of these results (not shown in this paper) the 100 and 500 experiments outperform the rest. However, the 100 and 500 experiments have a higher variance in the success rates of different seeds than the 10 and 50 experiments. High variance in the test results of the 100/500 RGB-D experiments shows that some seeds are able to achieve a near perfect score on the testing environment and others completely fail, thus there is then no empirical guarantee that RL agents will generalize when exposed to numerous environments.

The average success rate for the RGB input without the depth shows that domain randomization alone might not be an effective method to adapt the policy to variations in the observations, at least not in the context of RL. In fact, it shows little progress, e.g., the RGB agent exposed to one environment achieves around a 20% success on the testing environments and the agents exposed to 50+ environments achieve less than 40% success. These results are consistent when running with a grayscale channel (see Table 1).

While training by randomizing the environment did show some success in making supervised learning models generalize better, it fails to do so in RL policies. It is clear from these results, that adding random variations and relying solely on the RL objective is not enough to ensure generalization. Much of the success of domain randomization in previous works (Tobin et al., 2017) was reported using supervised learning. Also, the generalization abilities of machine learning algorithms have been linked to supervised learning setups. Therefore, it would make sense to adapt supervised learning techniques to regularize the models trained with DRL.

## 6.2. Invariance Regularization Experiments

In this section we will discuss the results obtained from training the agent using the method proposed in Section 5. As mentioned in Section 5, we propose two methods of using the proposed IR penalty. The first is to add to the PPO objective as in Equation 1, this method is referred to as *(full objective)* in the results. The second, which is referred to as *(split)* in the results, is to split the objective into two parts: The first part consists of training RL on the observations of the original training environment, while the second part can be seen as a supervised learning objective on the transformed observations, as shown in Algorithm 1. In the following experiments, in the split version, the model is trained with one iteration of the algorithm. Therefore, the training process has two stages, train RL then train with a supervised learning setup, without iterating between both.

As for the nature of transformation $\mathcal{T}$ of the observations, we tested with the same randomly textured environments from VizDoom, that were used in the previous section, in order to be able to make fair comparisons with the pure RL and domain randomization agents. In the split method from Algorithm 1, we train RL on one environment and then use the actions that the trained policy would have taken in that environment to tune the model with supervised learning on the textured environments. Regarding the distance penalty term $d$ in Equation 1, we did preliminary experiments with the KL divergence, $L1$, $L2$ and cross-entropy losses and the KL divergence returned the best results. Table 1 shows the results for combining PPO with the IR penalty using the two proposed implementations.

Observing the *split* method's results, we see that the success rates for the RGB-D input is close to that of the vanilla PPO agents. However, we see noticeable improvement in the 100/500 agents. The success across different seeds is more consistent and therefore the average performance is better and the variance is lower. Given RGB inputs our agent outperforms the agent trained just on domain randomization and RL. This shows the clear benefit of adding the IR term for the generalization of the learned policies. We also notice similar results for the Grayscale input which further shows that this method is helping the policy generalize even when the input does not contain explicit invariant channels.

Training with the *full objective*, however, returned the best results that outperform vanilla PPO with domain randomization and the split version of the IR algorithm. Similar to the split version, training on the full objective shows stable performance for the different inputs across different seeds. The results, in Table 1, also show that the trained models are achieving test success rate, with only 10

| Num training envs: | 1 | 10 | 50 | 100 | 500 |
|---|---|---|---|---|---|
| | | | *PPO* | | |
| RGB | $0.21 \pm 0.04$ | $0.17 \pm 0.04$ | $0.35 \pm 0.13$ | $0.35 \pm 0.16$ | $0.34 \pm 0.14$ |
| RGB-D | $0.05 \pm 0.04$ | $0.89 \pm 0.05$ | $0.90 \pm 0.05$ | $0.61 \pm 0.37$ | $0.77 \pm 0.33$ |
| Grayscale | $0.36 \pm 0.04$ | $0.33 \pm 0.13$ | $0.37 \pm 0.04$ | $0.47 \pm 0.14$ | $0.41 \pm 0.22$ |
| | | | *PPO-IR (split)* | | |
| RGB | - | $0.64 \pm 0.05$ | $0.69 \pm 0.03$ | $0.72 \pm 0.016$ | $0.75 \pm 0.02$ |
| RGB-D | - | $0.85 \pm 0.02$ | $0.90 \pm 0.05$ | $0.94 \pm 0.01$ | $0.95 \pm 0.02$ |
| Grayscale | - | $0.69 \pm 0.01$ | $0.76 \pm 0.02$ | $0.75 \pm 0.02$ | $0.76 \pm 0.02$ |
| | | | *PPO-IR (full objective)* | | |
| RGB | - | $\mathbf{0.79 \pm 0.05}$ | $\mathbf{0.79 \pm 0.03}$ | $\mathbf{0.81 \pm 0.03}$ | $\mathbf{0.81 \pm 0.02}$ |
| RGB-D | - | $\mathbf{0.98 \pm 0.01}$ | $\mathbf{0.97 \pm 0.01}$ | $\mathbf{0.99 \pm 0.01}$ | $\mathbf{0.99 \pm 0.01}$ |
| Grayscale | - | $\mathbf{0.79 \pm 0.03}$ | $\mathbf{0.79 \pm 0.01}$ | $\mathbf{0.79 \pm 0.02}$ | $\mathbf{0.80 \pm 0.02}$ |

*Table 1.* Average success rate and standard deviation of agents, that are trained on a different number of randomly environments, when tested on 50 test environments whose textures are not seen during training. The bold values represent the algorithm that resulted in the best average success rate according to an amount of training environments and an input type. We see that our method brings stability to the average results and improves generalization even when no depth is added.

training environments, that is close if not identical to the agents trained on 50, 100 and 500 environments. These results suggests that training with the full objective version of the IR algorithms does not require a large number of environments to learn the invariant features. Notice the average testing success rate is similar across the different number of training environments since the model learns the invariant features from only 10 environments and adding more environments that share the same invariant features will not make a difference. We can verify that hypothesis when looking at the RGB-D testing results in the full objective part. All agents achieve a near perfect score which we attribute to the availability of an invariant feature map in the input (the depth channel) which only the agents trained with the full objective are able to catch.

### 6.2.1. COMPARISONS WITH OTHER REGULARIZATION TECHNIQUES

Regularization has been shown to help in the generalization of supervised learning models (Srivastava et al., 2014). Using regularization in supervised learning often improves the performance of the trained models on test sets. However, regularization has not been frequently used in DRL setups, possibly due to the previously-common poor practise of testing and training in the same environment so there is no generalization gap (Cobbe et al., 2018).

We compare our method with some regularization techniques that are frequently used; dropout, batchnorm and $L2$. The first experiment has a dropout layer added after each convolutional layer (Srivastava et al., 2014), the second has a batchnorm layer added after every convolutional layer (Ioffe & Szegedy, 2015) and the last uses $L2$ regularization.

We choose the dropout probability to be 0.1 and the $L2$ weight to be $10^{-4}$, the same values that were proposed by Cobbe et al. (2018). As in the previous setup, we train five models (different seeds) for each technique and evaluate on 50 environments whose textures are sampled from a hold-out set. We report the experiments done with RGB input only as it poses a harder problem and a larger gap than RGB-D.

Figure 2 (left) shows the average success rate over 5 seeds for the four methods. We see that our proposed method is the only one that is steadily improving when more environments are added. The batchnorm models performed worst while dropout and $L2$ achieved similar success rates to the split version of our method given 50 and 500 training environments. However, the entropy of the learned policies is substantially higher when dropout and $L2$ are added to the model.

We hypothesize that the high entropy policies are able to generalize by acting randomly in some instances and this makes them more robust in certain situations. We show the *success weighted shortest path length* (*SPL*) in Figure 2 (right). SPL was proposed by Anderson et al. (2018) as a way of measuring the navigation agents success rates while taking into account the time it takes agents to succeed. A random behavior that displays robustness (has a high success probability) would return a relatively lower SPL due to the fact that this random behavior will probably

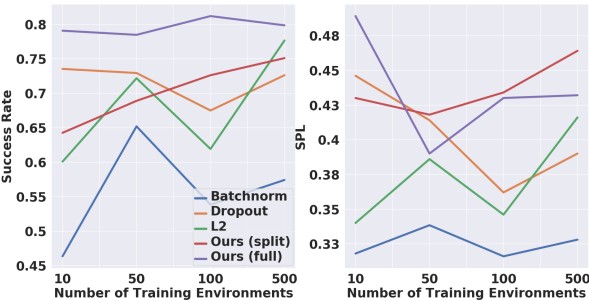

*Figure 2.* Different regularization methods tested on the RGB input. (Left): The average success rate show that some of these methods achieve similar results to ours in some instances. (Right): The lower SPL of the other regularization methods relative to ours indicates some randomness in their learned policies

not take the shortest possible path to the goal [1].

$$SPL = \frac{1}{N} \sum_{i=1}^{N} S_i \frac{l_i}{p_i}, \qquad (2)$$

where $N$ is the number of runs, $S_i$ is the binary indicator of the success of episode $i$, $l_i$ is the length of the shortest possible path and $p_i$ is the length of the path taken by the agent. Figure 2 (right) shows that the dropout and L2 agents have a lower SPL than the IR agents indicating that these policies with higher entropy are inefficient.

## 7. Discussion and Conclusions

We present a study of the generalization capabilities of visual navigation agents trained with deep reinforcement learning algorithms. We formalize what it means to generalize in the context of a POMDP. We reach a similar conclusion as in Cobbe et al. (2018) and Zhang et al. (2018), where the tendency of RL agent to overfit even when exposed to large training sets is quite visible. We show that using domain randomization with RL, without adding invariant features to the input such as the depth maps, is not enough to generalize. Even with added invariance, the agents showed high variance in the success rate on the testing environments across different seeds. In the second part, we proposed Invariance Regularization (IR), a method that combines supervised learning with RL to leverage the generalization ability of supervised learning techniques and regularizes the RL model. This algorithm improves the generalization success even with no added depth and displays stable performance across different seeds.

In this work, we focused our experimentation on

---

[1] The SPL formula suggested by Anderson et al. (2018) contains a $max(p_i, l_i)$ at the denominator instead of $p_i$, as written in Equation 2. We removed this factor as the max is redundant since $p_i \geq l_i$.

generalization to changes in the input observation because for visual navigation agents deployed in the real-world it would be very difficult to guarantee a stationary observation set. However, it is also interesting to generalize the learned skills to different architectural designs of the environment, just as one one wishes to generalize to different levels of the game as proposed in the retro competition (Nichol et al., 2018).

Another avenue of future work is to explore the appropriate transformation function $\mathcal{T}$ of the observations, that it would be useful to expose the agent to in order to help in robustness and generalization to other environments. One might consider generating adversarial examples as a way of improving generalization (Goodfellow et al., 2015). Another interesting research discussion would be to use an adaptive form of $\mathcal{T}$ coupled with data augmentation techniques. One might learn a model to find the augmentation strategy that makes the task harder on $\pi$ which may help it find a generalized representation, as done by Cubuk et al. (2018).

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
