# OpenReview forum: "Improving the Generalization of Visual Navigation Policies using Invariance Regularization"
_ICML.cc/2019/Workshop/RL4RealLife — RL4RealLife 2019_

### Official Review · AnonReviewer1 · 2019-05-25
**Good motivation and reasonable idea. Some concerns about the experiments.**

**Rating:** 4
**Confidence:** 4

**Review:**

The paper focuses on the generalization ability of visual navigation agents trained through deep RL. The authors argue that such agents are often trained and tested in the same environment, which cannot guarantee the ability of the agents to adapt to fresh real-world environments. The authors propose an invariance regularization method which introduces supervised learning with reinforcement learning to improve their generalization ability. Extensive experiments are conducted to verify the effectiveness of the proposed method.

Strengths:
1. The paper tackles an important and practical problem -- enhance the generalization ability of RL agents. The motivation is clear and convincing.

2. Most parts of the paper are well-written and easy to follow. The proposed method is intuitionally reasonable and easy to understand.

3. Elaborate experiments are conducted. They first show the classical domain randomization method is inefficient to learn the implicit invariant features for generalization, and then demonstrate the effectiveness of the proposed method.

Weaknesses:
1. The description of the experiment details in the 2nd paragraph of section 6.1 is not very clear. I’m not sure whether the agents are trained on the same number of episodes no matter how many environments are used? In the line 283 to 286, the authors mention that the performances of the agents trained on 100 and 500 environments are worse than those trained on 10 and 50, which is caused by overfitting. It seems the agents are trained on more episodes when there are more environments. But I think we need restrict the number of episodes to be the same for all the experiment settings to compare the ability of generalization.

2. Although the experiments on the inputs without depth channel demonstrate that the proposed method enhances the generalization ability by keeping some implicit invariant features, the proposed method doesn’t obviously beat the vanilla method when depth channel is introduced.

3. According to the 1st paragraph of section 6.2, only one iteration is performed to train the agents in the experiments, which is not consistent with the presented algorithm.

---

### Decision · Program_Chairs · 2019-05-28

Accept